# The Inequities of National Adaptation to Climate Change

Heidi K. Edmonds, C. A. Knox Lovell * and Julie E. Lovell

Centre for Efficiency and Productivity Analysis (CEPA), School of Economics, University of Queensland, Brisbane, QLD 4072, Australia

* Correspondence: knox.scholar@gmail.com

**Abstract:** With global efforts to mitigate climate change lagging behind what is necessary to achieve Paris Agreement global warming targets, global mean temperatures are increasing, and weather extremes are becoming more frequent and more severe. When mitigation falters, adaptation to current and anticipated future climate conditions becomes increasingly urgent. This study provides a novel collection of adaptive capacity and adaptation readiness indicators, which it aggregates into a composite adaptation index to assess the relative adaptation performance of nations. Adaptation performance is assessed using two complementary techniques, a distance to frontier analysis and a dominance analysis. Developed countries perform relatively well and developing countries perform relatively poorly in both exercises. Adaptation performance is found to be closely and positively related to both national income per capita and greenhouse gas emissions per capita, highlighting the inequities of global adaptation performance. These adaptation inequities are consistent with the IPCC assessment that nations most affected by climate change are those that are least able to adapt and contribute least to the problem, creating a need for assistance from developed countries.

**Keywords:** climate change; adaptive capacity indicators; adaptation readiness indicators; composite adaptation index; adaptation inequity

## 1. Introduction

The 2015 Paris Agreement sought to limit global warming to a 2 °C increase above pre-industrial levels by the end of the century, with an aspirational increase of 1.5 °C. The Intergovernmental Panel on Climate Change (IPCC) [1] has predicted a greater than 50% likelihood that global warming will reach or exceed 1.5 °C much sooner, between 2030 and 2052, even for a greenhouse gas emissions scenario that increases at less than the current rate. In its Adaptation Gap Report 2021 launched at COP26, the United Nations Environment Programme (UNEP) [2] warned that mitigation efforts to cut greenhouse gas emissions are " . . . still not anywhere near strong enough . . . ", with countries currently on track to experiencing a 2.7 °C increase. UNEP [3] predicted that implementation of current mitigation pledges implied a 50% chance of keeping global warming to 2.5 °C (with a range of 2.0 °C to 2.9 °C) by the end of the century.

An editorial in Nature Climate Change [4] warned that mitigation actions and pledges to reduce greenhouse gas emissions have been insufficient to meet either Paris Agreement target, making adaptation increasingly urgent. The IPCC [5] Sixth Assessment Report (AR6 WG II) emphasised the significance of adaptation by noting that near-term actions to mitigate emissions that would limit global warming would reduce projected losses and damages in both human systems and ecosystems but would not eliminate them. As in previous Assessment Reports, it stressed the inequity of the distribution of the adverse impacts of global warming, with the most vulnerable people and systems being disproportionately affected. Unlike previous Assessment Reports, it highlighted adaptation actions that are effective, equitable, and just. These views echo those expressed so eloquently in The Stern Review [6]. Stern argued that the climate will continue changing through the near future, however successful mitigation efforts are, and without early and strong mitigation the costs

of adaptation will rise exponentially. Moreover, the communities affected first and most severely are often those that are least able to adapt and contribute little to the problem, creating a double inequity and warranting increased assistance from developed countries. This study examines the effectiveness and, following Stern, the equity of adaptation to climate change.

A range of constraints can prevent adaptation from reaching its potential. Some nations adapt better than others, perhaps because they face fewer constraints to adaptation than others, inadequate resourcing and financing being prominent constraints, limited governance competence another. Adger and Barnett [7] and Massetti and Mendelsohn [8] claimed that while adaptation has the potential to greatly reduce climate change damage, there is no guarantee this potential will be reached. The widely proclaimed distinction between adaptive capacity and adaptation suggests an application of the concepts of best practice and dominance to evaluate nations' adaptation performance. Analytical techniques have been developed to implement the concepts of best practice and dominance, with empirical performance evaluation applications ranging widely, from hospitals and schools to industries and nations, and even to the environmental performance of businesses (Trinks et al., [9]) and nations (Bosetti and Buchner [10], Matsumoto et al., [11]). This study applies these concepts to assess nations' climate change adaptation performance.

Adaptation, particularly transformative adaptation, requires resources and financing often unavailable in developing countries. UNEP [2] noted that while mitigation is the preferred way to lower the impacts and costs of climate change, and recent climate change financing has gone primarily to mitigation, support for adaptation is critical to keep existing adaptation gaps between implemented adaptation and societally set adaptation goals from widening. It documented that planning, financing, and implementation of adaptation remain weak, and only a small portion of the fiscal stimulus to combat the COVID-19 pandemic has targeted climate finance, and a small portion of that has gone to adaptation. IPCC [5] AR6 WG II documented that recent adaptation progress has been distributed unevenly across regions, with observed adaptation focused more on planning than implementation. In his statement on the release of IPCC [12], United Nations Secretary-General António Guterres called adaptation "the neglected half of the climate equation" and urged the public and private sectors to work together to ensure a just and rapid transformation to a net zero emissions global economy.

The potential gains to adaptation can be substantial. The Global Commission on Adaptation [13] has estimated that a $1.8 trillion investment in adaptation, including early warning systems, climate-resilient infrastructure, improved agricultural practices, mangrove protection along coastlines, and resilient water resources, could generate $7.1 trillion in benefits through a combination of avoided costs and a variety of social and environmental benefits. It also estimated that universal access to early warning systems could deliver benefits up to ten times the initial cost.

With this background, the study has three objectives. The first is to obtain a set of adaptive capacity and adaptation readiness indicators and to aggregate these indicators into a composite adaptation index for a large number of nations; this appears in Sections 2–4, with guidance from the IPCC and related literatures. The second is to provide an analytical framework for modelling and quantifying the composite adaptation performance of nations; this is the subject of Section 5, in which a pair of complementary performance evaluation techniques is developed. The third and most significant objective is to explore the distribution of the composite adaptation performance of nations, and to illustrate its double inequity; this is the subject of the empirical analysis in Section 6.

The paper proceeds as follows. Sections 2 and 3 survey the literature seeking to define adaptive capacity and adaptation readiness, the activities that contribute to them and the factors that constrain them, and the inequity of their distributions. Section 4 introduces and evaluates the data used to assess the adaptation performance of nations. These data are obtained from the University of Notre Dame Global Adaptation Initiative [14] and are used in a manner guided by the IPCC and related literatures. Section 5 describes the distance to

frontier and dominance techniques used to implement an assessment of nations' adaptation performance. Section 6 presents the empirical findings of the study, which provide strong support for the growing emphasis the IPCC has placed in successive Assessment Reports, and strongly urged by Secretary-General Guterres at COP26, for the need to increase climate change funding, and to reallocate funds from mitigation to adaptation and from developed to developing nations. Section 7 concludes with a summary of the findings and their implications for the climate justice movement, and notes two limitations of the study that may spur additional research into climate change adaptation.

## 2. The IPCC on Adaptation

The IPCC [15] Third Assessment Report AR3 of Working Group II (AR3 WG II) expressed five reasons for concern about vulnerability to climate impacts, concerns which it has re-evaluated in subsequent Assessment Reports. The reasons involve relationships between global mean temperature increase and five projected adverse impacts: risk of damage to or irreparable loss of unique and threatened systems, the distribution of its impacts, the magnitude of aggregate impacts, the risk of extreme weather events, and the risk of large-scale singular events. These concerns led it to consider the role of adaptation, which it defined as adjustment in natural and human systems to the actual or expected impacts and risks of climate change, and it noted that adaptive capacity is more limited in natural systems than in human systems. It distinguished adaptation from adaptive capacity, which it defined as the ability (emphasis added) of a system to adjust to climate change, to moderate potential damages, or to cope with the consequences. It viewed adaptation as a necessary strategy to complement climate change mitigation efforts and noted that the ability to adapt to and cope with climate change depends on wealth, technology, education, information, human and social capital, infrastructure, institutions, management capabilities, and access to resources. Subsequent Assessment Reports have expanded on these drivers, emphasising the roles of technology, citing new and possibly disruptive technologies and enhanced climate-driven innovation, and a supportive institutional framework.

AR3 WG II attributed the difference between actual adaptation and adaptive capacity to maladaptation and constraints to achieving potential adaptation. This distinction was strengthened in IPCC [16] AR4 WG II, which stated "The message from the literature is clear: adaptive capacity signals potential but does not guarantee adaptive action", and warned that more extensive adaptation than was currently occurring would be required to reduce vulnerability to future impacts of global warming. It cited regulations and policies, limited governance capacity, availability and distribution of finance, violent conflict, the spread of infectious diseases, and urbanisation as factors that may facilitate or constrain adaptation. This list of constraints has changed little through successive Assessment Reports.

Of particular relevance to this study, the IPCC [15] AR 3 WG II asserted that nations with the least resources have the least capacity to adapt and are most vulnerable to climate change impacts. It also asserted that the projected distribution of impacts would increase the disparity in well-being between developed and developing nations, with the disparity growing for higher projected temperature increases. The IPCC [17] AR5 WG II noted that differences in vulnerability and exposure that require adaptation arise from non-climatic factors and from multidimensional inequalities associated with uneven development processes, with people who are socially, economically, culturally, politically, institutionally, or otherwise marginalised being especially vulnerable. The IPCC [1] Report on Global Warming of 1.5 °C identified populations and regions at disproportionately higher risk of adverse impacts, including disadvantaged and vulnerable populations, local communities dependent on agricultural or coastal livelihoods, small island developing states (SIDS), and Least Developed Countries (LDCs). It suggested that social justice and equity are core aspects of climate-resilient pathways, that a consideration of ethics and equity can address the uneven distribution of adverse impacts and concluded that international cooperation is critical for developing countries and vulnerable regions. The IPCC [5] AR6 WG II stated that poor and otherwise disadvantaged groups are especially vulnerable because they have

fewer assets and less access to funding, technologies, and political influence. People in the most vulnerable situations and regions are also highly exposed to climate change impacts. The most vulnerable regions include South Asia, Micronesia and Melanesia, Central America, and most of Africa.

These definitions of adaptation, adaptive capacity, and adaptation constraints, and the proclaimed inequity in the distribution of adaptive capacity, have changed little through successive Assessment Reports. A second consistency is the lack of a definition of adaptation readiness, although some of its drivers appear often (e.g., innovation, education, governance) (The IPCC [5] AR6 WG II has introduced the concept and extolled the significance of enabling conditions, including political commitment, institutional frameworks, knowledge, and monitoring and evaluation. These conditions overlap with the adaptation readiness indicators used in this study.) A third consistency involves the growing acknowledgement of an association between the greenhouse gas emissions of developed countries and the climate change impacts that disproportionately affect developing countries responsible for few emissions. What has changed in successive Reports is the knowledge base, consisting of advances in science and increases in the quantity and quality of evidence in databases and in the scientific, technical, and socioeconomic literature. This has allowed the IPCC to increase the confidence it attaches to its assessments of the relationship between global warming and its impacts in each successive Assessment Report.

## 3. Academia on Adaptation

There has been much cross-fertilisation between the IPCC and academe, with the IPCC inspiring research in academe, whose findings provide evidence that influences the confidence the IPCC attaches to its conclusions. This Section explores four frequently intersecting topics from a voluminous primarily academic literature relevant to this study. (Not all cited sources originate in academe, but they influence academic writings).

### 3.1. The Role of Business

Businesses face growing pressure to adapt to climate change from environmental regulations, shareholder activism, and changing consumer demands. Recovery from the COVID-19 pandemic also has created environmentally friendly investment opportunities for businesses. Since business creates a large share of national output, business adaptation accounts for a similarly large share of national adaptation.

The United Nations Framework Convention on Climate Change (UNFCCC) [18] stated the business case for adaptation, listing five generic risks requiring adaptation: physical risk (e.g., damage to physical assets), price risk (e.g., price increases of raw materials or supply chain bottlenecks), regulation risk (e.g., new environmental regulations that raise operating costs), reputation risk (e.g., business reputations linked to their environmental impacts), and liability risk (e.g., investors seeking compensation for avoidable or uninsured losses attributable to climate change).

Raynor and Pankratz [19] argued that businesses respond to climate change through three dimensions: mitigation (e.g., reducing business emissions), adaptation (e.g., moderating harm to business operations by reducing exposure to climate-related risks), and value creation (e.g., creating products and services designed to exploit the beneficial opportunities presented by climate change, such as onshore wind turbines and utility-scale photovoltaics). Prioritising long-term sustainable profitability rather than quarterly growth and profit may make actions related to climate mitigation, adaptation, and value creation more justifiable in financial terms. Balaouras and Schiano [20] claimed that businesses are committing to ambitious climate action plans, including investments in decarbonisation, renewable energy procurement, infrastructure upgrades and retrofits, relocation and migration, supply chain resilience improvements, and weather proofing. Kahn [21] argued that adaptation through induced, directed innovation would power adaptation in all activities affected by climate change, including the scaling of wind and solar power, the introduction of dynamic electricity pricing, the development of improved weather prediction models,

and the development of small-scale insurance products in agriculture, which is particularly susceptible to climate change.

Global management consulting firm McKinsey and Company has been prominent in making the case for business adaptation to climate change. In one of a series of posts during COP26, McKinsey and Company [22] stressed the role of innovation and directed technical change, suggesting that nearly 40% of emissions abatement required to meet Paris Agreement targets could come from technologies that are either still in R&D or demonstrated but not yet mature. It cited green hydrogen as a prominent example, with likely applications to the carbon-intensive chemicals and steel industries. (See Acemoglu [23] and Acemoglu et al. [24] for details on directed technical change, especially as it applies to the environment.)

### 3.2. The Role of Government

Government plays three roles, providing an institutional environment supportive of private adaptation, providing adaptation goods and services having public good features that the private sector cannot provide efficiently, and directing public and private pandemic recovery investment in an environmentally sustainable direction.

The first role, providing an enabling environment for private adaptation activities, creates what we call adaptation readiness. (The term "adaptation readiness" has been interpreted in two ways in the literature. We follow Ford and King [25], Salamanca and Nguyen [26], Sarkodie and Strezkov [27], Amegavi et al. [28], and Adom and Amoani [29], in interpreting the term as describing the institutional environment within which private adaptation occurs. Tilleard and Ford [30] interpreted the term as combining the institutional environment with private adaptation activities.) An important component of the government's role is to lower barriers, or constraints, to private adaptation. The IPCC [15] classified barriers as financial, informational and cognitive, and social and cultural, to which Biesbroek et al., [31] added institutional. The academic literature, including Moser and Ekstrom [32], Fankhauser [33], Massetti and Mendelsohn [8] and many others, has fleshed out these classifications with numerous examples of constraints to private adaptation performance.

The second role for government is to exploit joint, or public, adaptation opportunities. Successive IPCC Assessment Reports have distinguished adaptation opportunities having a single beneficiary that can be undertaken by private agents from adaptation opportunities having many beneficiaries that are best undertaken by governments. Self-interest should elicit efficient private adaptation, but the public good features of joint adaptation require government intervention to achieve efficient joint adaptation. Fankhauser et al., [34], Mendelsohn [35,36], Hanemann [37], and Anderson et al. [38] have considered the roles of markets and governments in adaptation, by distinguishing private from joint adaptation. Examples of the latter include water supply, coastal protection, public health, weather forecasts, and ecosystem preservation.

There is widespread interest on the part of business and government in pursuing an environmentally sustainable recovery from the COVID-19 pandemic, the third role for government. The OECD [39] has consistently supported a low-carbon recovery. UNEP [2] observed that the pandemic has increased vulnerability to climate change and increased pre-existing financial barriers to investment in adaptation. It also noted the opportunity to reduce these barriers by directing recovery funding into green and resilient recoveries, but lamented that countries are missing the opportunity to use fiscal recovery to prioritise green economic growth that supports adaptation to climate change. It calculated that just a small portion of stimulus funding globally has targeted adaptation, despite an urgent need to increase public adaptation finance both for direct investment and for overcoming barriers to private adaptation. UNEP [3] pointed to the potential for public investment in climate change mitigation and adaptation to raise long-term prosperity by creating jobs, accelerating economic growth, and meeting environmental, gender and social objectives. Dr. Fatih Birol, Executive Director of the IEA, has urged governments and financial institutions

to stimulate recovery from the pandemic by funding investment in new technologies such as clean power, battery storage, and carbon capture technology. (Birol's recent comments can be found at https://www.iea.org/authors/dr-fatih-birol, accessed on 12 October 2022) The Global Center on Adaptation [40] reported that over 3000 scientists, including five Nobel Laureates, from over 100 nations signed the "Groningen Science Declaration" urging pandemic recovery programs to give priority to adaptation policies such as green job creation. Mérida [41] has argued that a combination of green hydrogen, renewable electricity, and digital technologies can revolutionise the global energy system. He cited the Hydrogen Council as estimating that by 2050 hydrogen will account for 18% of energy usage, avoid six gigatonnes of annual carbon emissions, and create 30 million jobs.

### 3.3. The Role of Health

Climate change and the pandemic have been linked to a host of health-related outcomes, which in turn impact on the adaptive capacity of nations.

Prior to COP26 the World Health Organization (WHO) [42] released a special report enumerating the health impacts of climate change, including impacts on healthcare facilities, and emphasised the need for health to occupy a prominent role in the climate change agenda. It argued that the pandemic and climate change have had a compounding impact on the adaptive capacity of governments and societies, with disproportionate health, economic and social impacts for those that are already vulnerable. UNEP [3] observed that the pandemic has increased global extreme poverty, the first increase in over 20 years, with pandemic recovery spending nearly USD 12,000 per capita in advanced economies and less than USD 60 per capita in low-income economies.

Increased disparities in public health are a direct consequence of increased poverty. In September 2021, just two months ahead of COP26, Atwoli et al., [43], representing editors of over 230 medical journals around the world, published a joint statement calling for urgent action to limit average global temperature increases to 1.5 °C above pre-industrial levels. The statement also urged leaders to restore biodiversity and protect public health, both of which global warming threatens, with harm disproportionately affecting the most vulnerable. The statement proposed a range of mitigation and adaptation plans and strategies to achieve this objective, headlined by replacing dirty technologies with clean technologies and the protection of public health care. One week ahead of COP26, Burki [44] reiterated the importance of the link between climate change and health, noting that only 0.3% of climate change adaptation funding is allocated to national health systems. He observed that national health ministries sought a commitment from COP26 to provide human and financial resources required to build climate-resilient and environmentally sustainable health systems. He concluded that the ethical case for acting on climate change is incontrovertible, and the economic case is just as strong.

### 3.4. The Role of Inequity

Successive IPCC Assessment Reports have noted the unequal distribution of vulnerability and adaptive capacity among and within nations and stated that climate change is exacerbating existing inequities.

Stern [6] argued that, historically, rich countries have produced the majority of greenhouse gas emissions and developing countries have suffered the consequences because of their geography, their dependence on agriculture, and their limited adaptive capacity. He continued by stating "There is therefore a double inequity in climate change: the rich countries have special responsibility for where the world is now, and thus for the consequences which flow from this difficult starting point, whereas poor countries will be particularly badly hit". Stern called on developed countries to honour their existing commitments to provide financial aid to developing countries to support their adaptation efforts.

Füssel [45] conducted an empirical test of Stern's double inequity hypothesis, by comparing the socio-economic capability and causal responsibility of nations on the one hand, and the vulnerability of nations in four climate-sensitive sectors, water supply, food security,

human health, and coastal zones and their populations on the other. He demonstrated the first inequity by showing that some nations have more adaptive capacity than other nations, as measured by their economic capability (gross domestic product (GDP) per capita) and their social capability (the United Nations Development Programme (UNDP) [46] Human Development Index). Additionally, nations with the most capability are most responsible for climate change since greenhouse gas emissions (fossil $CO_2$ emissions per capita cumulated since 1990) are highly correlated with these capability indicators. Füssel asserted that this double inequity " . . . strengthens the moral case for financial and technical assistance from those countries most responsible for climate change to those countries most vulnerable to its adverse impacts." Diffenbaugh and Burke [47] reached similar conclusions, finding that many poor countries have been significantly harmed by global warming arising from wealthy countries' energy consumption, either because they lack the resources to adapt, or because they are located in warmer regions where additional warming is detrimental to health and productivity. They found the ratio between top and bottom deciles of the population-weighted country-level per capita GDP distribution to be 25% larger than it would be without global warming. They concluded that since " . . . wealthy countries have been responsible for the vast majority of historical greenhouse gas emissions, any clear evidence of inequity in the impacts of the associated climate change raises critical questions of international justice".

Khan et al., [48] examined 25 years of adaptation finance justice, revisiting Stern's call for financial aid flows from developed to developing countries. They defined adaptation finance justice as raising adaptation funds according to the responsibility for climate impacts, and allocating funds putting the most vulnerable first, and concluded that climate justice has not been achieved, with a refusal by wealthy nations to define commitments in relation to responsibility and needs. Alcaraz et al. [49] proposed the opposite strategy, reallocating the global carbon budget consistent with a 2 °C global mean temperature increase using climate justice criteria. Simmons [50] and Klinsky [51] summarised the concept of climate change as a justice issue by arguing that key groups are affected differently by climate change, and by demonstrating that countries most vulnerable to climate change are the least responsible for generating the causal $CO_2$ emissions.

## 4. The Data

Many of the variables cited above as influencing adaptation to climate change appear in the ND-GAIN country data from the University of Notre Dame Global Adaptation Initiative [14]. These data are therefore used, in a manner guided by the IPCC and related literatures. The ND-GAIN country index is constructed from 36 vulnerability indicators and nine readiness indicators for up to 192 nations over varying time periods concluding in 2019. The vulnerability indicators consist of 12 exposure indicators, 12 sensitivity indicators and 12 adaptive capacity indicators, each measured on [0, 1] with low (high) values indicating low (high) vulnerability. (The ND-GAIN indicators have been transformed from raw data. The University of Notre Dame [14] provides raw data and derived indicators, and The University of Notre Dame [52] provides details of the transformation procedures. The indicators have been used often to study climate change vulnerability and adaptation; among recent studies are Edmonds et al. [53], Halkos et al. [54], Amegavi et al. [28], and Ripple et al. [55]).

The selection of data is guided by the observation that ND-GAIN defines adaptive capacity as "the ability of society and its supporting sectors to adjust to reduce potential damage and to respond to the negative consequences of climate events . . . ". The 12 adaptive capacity indicators " . . . seek to capture a collection of means, readily deployable to deal with sector-specific climate change impacts". ND-GAIN defines adaptation readiness as preparedness " . . . to make effective use of investments for adaptation actions thanks to a safe and efficient business environment . . . ", and it measures adaptation readiness with three components: economic readiness, governance readiness and social readiness. These interpretations and definitions suggest a strong complementarity between adaptive

capacity and adaptation readiness and are consistent with the views expressed in the IPCC and related literatures reviewed in Sections 2 and 3. They also support the creation of a composite adaptation index combining the two concepts, since adaptive capacity itself is insufficient for successful adaptation without the political, social, and institutional support provided by adaptation readiness. (Amegavi et al. [28] used a subset of the database we use to show that adaptation readiness is significantly and negatively related to vulnerability to climate change in 51 African nations. Our results support this finding and point to the significance of adaptation readiness in a larger sample of nations).

Consequently, it is hypothesised that the overall adaptation performance of nations is a function of their adaptive capacity and features of their institutional environment that enhance or constrain their adaptive capacity. These features are called enabling conditions in the IPCC [5] AR6 WG II and captured by the adaptation readiness indicators in this study. To test this hypothesis 12 adaptive capacity indicators and nine adaptation readiness indicators are extracted from the ND-GAIN database for the terminal year 2019. The adaptive capacity indicators are augmented with the adaptation readiness indicators because the IPCC and other literatures reviewed in Sections 2 and 3 consistently refer to various enabling conditions (e.g., regulatory quality, innovation, and education) as being important elements in the performance of nations to adapt to unmitigated climate change. Table 1 lists the ND-GAIN adaptive capacity and adaptation readiness indicators. Each of the 21 ND-GAIN indicators is designed to capture both capacity and access characteristics. Detailed descriptions of and rationale for each indicator appear in University of Notre Dame [52].

**Table 1.** ND-GAIN Indicators.

| Adaptive Capacity | Adaptation Readiness |
|:---:|:---:|
| Food | Economic Readiness |
| Agricultural Capacity | Ease of Doing Business |
| Child Malnutrition | |
| | Governance Readiness |
| Water | Political Stability and Non-Violence |
| Dam Capacity | Control of Corruption |
| Access to Reliable Drinking Water | Rule of Law |
| | Regulatory Quality |
| Health | |
| Medical Staffs | Social Readiness |
| Access to Improved Sanitation | Social Inequality |
| | Information and Communication |
| | Technology Infrastructure |
| Ecosystem Services | Education |
| Protected Biomes | Innovation |
| Engagement in International | |
| Environmental Conventions | |
| | |
| Human Habitat | |
| Quality of Trade and Transport- | |
| Related Infrastructure | |
| Paved Roads | |
| | |
| Infrastructure | |
| Electricity Access | |
| Disaster Preparedness | |

Three adjustments have been made to the ND-GAIN data. The 12 adaptive capacity indicators have been transformed because ND-GAIN associates high vulnerability indicators with high vulnerability, and adaptive capacity is one of three components of vulnerability. Since adaptive capacity reduces vulnerability, each adaptive capacity indicator is redefined so that high values of each adaptive capacity indicator are associated

with high adaptive capacity, thereby retaining their [0, 1] range. Two of the transformed ND-GAIN adaptive capacity indicators, "improved water source (% of population with access)" and "improved sanitation facilities (% of population with access)" have missing values for 94 and 103 nations, respectively, and these indicators have been deleted, leaving 10 adaptive capacity indicators. A new water indicator was adopted, the geometric mean of "dam capacity" from ND-GAIN and "average precipitation in depth (mm per year)" from the World Bank's World Development Indicators. This new water indicator combines rainfall with water storage capacity and provides a nearly necessary condition for the original water indicator "improved water source (% of population with access)", while greatly increasing coverage from 98 to 134 nations, leaving 11 adaptive capacity indicators.

These three adjustments generate a pair of data matrices, one consisting of 11 adaptive capacity indicators for up to 192 nations, and the other consisting of nine adaptation readiness indicators for up to 192 nations. However, these two matrices contain many missing observations. One adaptive capacity indicator, disaster preparedness, is missing for 56 nations, and two adaptation readiness indicators, social inequality and innovation, are missing for 43 and 44 nations. These three indicators have been deleted, leaving nine adaptive capacity indicators and seven adaptation readiness indicators. If a nation is missing one or more of the nine remaining adaptive capacity indicators, that nation is deleted from the adaptive capacity matrix, and similarly for the adaptation readiness matrix. This leaves an adaptive capacity matrix consisting of nine indicators for 143 nations and an adaptation readiness matrix consisting of seven indicators for 172 nations. In order to merge information on adaptive capacity with information on adaptation readiness into a composite adaptation index, the sample is restricted to nations having values for all 16 indicators. This leaves an adaptive capacity matrix consisting of nine indicators and an adaptation readiness matrix consisting of seven indicators, both for the same 134 nations. Summary statistics for the 16 indicators appear in Appendix A Table A1.

These two data matrices reflect the difficult trade-off between coherence and comprehensiveness of indicators and comparability and inclusiveness of nations. (For a conceptual treatment of this trade-off, see Ford and Berrang-Ford [56], who proposed four requirements for successful adaptation tracking: (1) a consistent definition for monitoring panel data, (2) observed units must be comparable, (3) sample size must be large enough to be comprehensive, and (4) indicators must be coherent with our understanding of adaptation.) A preference for inclusiveness reflects our desire to retain as many developing nations as possible. The data set contains 32 Least Developed Countries (LDCs) identified by the UN and 11 SIDS identified by the UN and includes 27 sub-Saharan African nations and 10 North African nations. These nations have been singled out by the IPCC and at COP26 as being victims of climate change caused largely by developed nations, who have lagged in both their mitigation efforts and their financial support to developing nations to enhance their adaptation performance through National Determined Contributions. This dichotomy has been labelled an equity and ethical issue in consecutive IPCC Assessment Reports, and a justice issue by many, including by Robinson and Shine [57], Simmons [50], and Klinsky [51].

## 5. The Analytical Techniques

A pair of complementary analytical techniques are used to assess the relative adaptive capacity and adaptation readiness of nations. Each technique is illustrated using adaptive capacity, and the same analysis applies to adaptation readiness and composite adaptation. Both techniques identify leading and lagging nations. The first identifies leaders and laggards by using Data Envelopment Analysis (DEA) to exploit the "distance to frontier" concept of Acemoglu et al., [58] and applied to OECD productivity dispersion by Andrews et al. [59] and Berlingieri et al. [60,61]. The second identifies leaders and laggards by using dominance analysis to identify nations that are structurally similar but perform better than other nations, regardless of their distance to the best practice frontier.

The first technique, DEA, is a linear programming technique developed by Charnes et al. [62] to assess the relative performance of observations in a sample. Rather than fitting a regression through the data, as most statistical techniques do, DEA constructs a frontier that envelops the data, from above if the objective is to maximise and from below if the objective is to minimise. The frontier consists of best practice observations, and with a maximisation orientation, all observations beneath the frontier lag best practice by varying degrees. In the current setting higher adaptation indicator values are preferred, and DEA constructs an adaptive capacity frontier that bounds an adaptive capacity set from above. The adaptive capacity frontier consists of best practice nations, those that adapt best, and the interior of the adaptation set contains all nations whose adaptation performance lags best practice, or the "best" and the "rest" in the OECD productivity literature. DEA simultaneously identifies adaptation leaders on the best practice frontier and measures the radial distance to the frontier of the adaptation laggards. Distance to the frontier provides a new measure of the adaptation gap.

Let nations be indexed by i = 1, . . . , I, and let a nation's adaptive capacity be tracked across N sectors with sectoral adaptive capacity indicators labelled $y_n$ and indexed by n = 1, . . . , N. In the current application I = 134 and N = 9. The DEA program that evaluates the aggregate adaptive capacity to cope with climate change of nation "o" is given by the dual pair of linear programs in Figure 1. These programs calculate an endogenously weighted adaptive capacity index ACI for each nation. This index aggregates N individual adaptive capacity indicators $y_n$ into a single adaptive capacity index ACI.

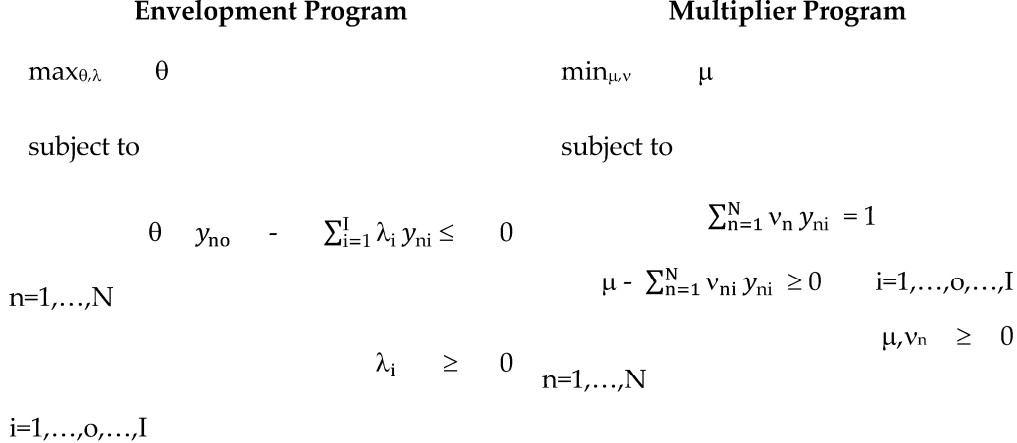

**Figure 1.** DEA ACI Programs.

The envelopment and multiplier programs contain sectoral adaptive capacity indicators $y_n$ but no additional variables that might influence adaptive capacity such as resource availability. This abbreviation of a conventional DEA program is the contribution of Adolphson et al., [63], and Lovell and Pastor [64]. Unlike most models of business or economic behaviour that contain variables to be maximised, such as business revenues or educational outcomes, and constraining variables, such as business expenses or resource availability, this adaptive capacity model is restricted to variables to be maximised, the N sectoral adaptive capacity indicators. The envelopment program in Figure 1 envelops nations' adaptive capacity data from above and calculates the potential of nation "o" to expand its vector of sectoral adaptive capacity indicators $y_o$ as much as possible, subject to N constraints, one for each sectoral indicator. These constraints bound the expanded vector $\theta\, y_o$ above by a nonnegative combination of the most capable nations in the sample.

The optimal value of $\theta \in [1, +\infty)$. A value $\theta = 1$ indicates best practice adaptation on the adaptive capacity frontier, with larger values of $\theta$ indicating the degree to which a

nation must improve its adaptation performance to reach the best practice frontier. $\theta$ also forms the basis for a measure of a nation's adaptive capacity gap, the difference between (or ratio of) its actual adaptive capacity $y_o$ and its potential adaptive capacity $\theta\, y_o$. Deviations beneath this frontier capture an alternative representation of nations' adaptation gaps to the UNEP Adaptation Gap Reports of the same name by replacing a vague "societally set goal" with a best practice that can be estimated empirically.

The reciprocal $\theta^{-1} \in (0, 1]$ is a nation's adaptive capacity index ACI. A value $\theta = 1$ indicates best practice adaptation, and lower values of $\theta^{-1}$ indicating reduced levels of adaptive capacity. $\theta^{-1}$ provides a ranking of nations based on their overall adaptive capacity to cope with climate change, independently of any other national characteristics, which are ignored in the present analysis.

The multiplier program in Figure 1 calculates for nation "o" a vector of endogenous weights $v_n \in (0, +\infty)$ with which to aggregate its sectoral adaptive capacity indicators into its ACI. By the duality theorem of linear programming, at optimum $\theta = \mu$, and ACI can therefore be expressed as an endogenously weighted sum of its sectoral adaptive capacity indicators, $\theta^{-1} = (\sum_{n=1}^{N} v_n x_{ni})^{-1}$ for any nation $i = 1, \ldots, o, \ldots, I$. (These endogenous weights are also known as "benefit of the doubt" weights, a term introduced by Melyn and Meusen [65]. Cherchye et al. [66] provide details on benefit of the doubt composite indices.)

Endogeneity of weights is central to the analysis, having the virtue of not forcing nations to value sectoral adaptive capacities equally. For a nation with an abundance of sectoral adaptive capacity indicator $y_n$ the program implicitly attaches a large weight to this indicator to maximise its ACI. Conversely, for a nation with a paucity of sectoral adaptive capacity indicator $y_n$ the program implicitly attaches a small weight to this indicator to maximise its ACI. These endogenous weights provide a considerable improvement over the fixed weights used in most composite indices, including the popular UNDP [46] Human Development Index and the ND-GAIN indices. Fixed weights impose perfect substitutability among component indicators, with rates of substitution constant across nations. The endogenous weights generated by DEA also impose perfect substitutability among component indicators, but with the important advantage that weights, and rates of substitution among component indicators, are allowed to differ across nations according to their circumstances. Weight flexibility is particularly important in the construction of an adaptive capacity index, since nations differ in their exposure, sensitivity, and vulnerability to climate change across sectors. Endogeneity of weights allows Pacific Island nations to value adaptation indicators differently than sub-Saharan African nations. New Zealand has ample rainfall, and Mauritania is arid, leading to the expectation that New Zealand assigns a relatively high weight and Mauritania assigns a relatively low weight to a water indicator. By reflecting different adaptive capacities across sectors that in turn reflect different national circumstances, these weights have the potential to assist in the design of policies intended to allocate climate finance to enhance adaptive capacity in an equitable manner, as noted in Section 1 with reference to Mendelsohn [36] and Anderson et al. [38].

However, endogeneity of weights has a potential drawback. As the number of choice variables relative to the sample size increases, estimation becomes exponentially more difficult, a situation referred to as the curse of dimensionality. In our setting the number of adaptive capacity indicators relative to the number of nations in the sample $N/I = 9/134$ is sufficiently large to hinder evaluation of the adaptation performance of nations. In effect, having nine adaptive capacity indicators gives nations excessive freedom to choose weights in creating their ACIs, resulting in many nations receiving ACI = 1, even though their index is the consequence of being different rather than excelling. The curse is less severe in the case of adaptation readiness, where $N/I = 7/134$.

Summarising, the DEA methodology makes three contributions to the construction of an adaptive capacity index. It exploits the ability to generate endogenous weights with which to aggregate sectoral indicators that respect nations' varying circumstances. Nations' endogenously weighted adaptive capacity indices provide an analytically sound way of identifying leaders and laggards and quantifying adaptation gaps. These endogenous

weights have the potential to guide policy intended to lower the cost of enhancing adaptive capacity in an efficient, i.e., resource-saving, and equitable manner.

The second technique, dominance analysis, provides information complementary to that provided by DEA. The basics of dominance analysis are extracted from a much more detailed presentation in Tulkens [67]. Consider two nations with adaptive capacity vectors $y_j$ and $y_k$. Nation j dominates nation k if nation j has at least as much adaptive capacity as nation k for all N indicators, that is if $y_{nj} \geq y_{nk}$, $\forall$ n = 1, ... , N. Aggregating the inequality over all k = 1, ... , I nations generates the number of nations nation j dominates. Reversing the inequality generates the number of nations that dominate nation j. This strategy can be extended by deleting d $\geq$ 1 adaptive capacity indicators at a time, with replacement, to evaluate dominance with N-d indicators. This provides a way of determining the indicators for which a nation is most or least dominant.

Dominance analysis is independent of the notions of best practice adaptation and an adaptive capacity index. Rather, it identifies leaders as the most frequently dominating nations and laggards as the most frequently dominated nations. In doing so it identifies role models for dominated nations. These role model nations are relevant because they have similar mixes of adaptive capacity indicators, but with larger indicator values. It is important to note that a nation can dominate other nations by being similar to them and without being best practice, and a nation can be best practice by being different from other nations and without dominating any of them. This distinguishes dominance analysis from DEA and highlights the complementarity between the two techniques.

This exposition of DEA and dominance analysis has been illustrated with application to adaptive capacity, and the analysis applies equally to adaptation readiness and composite adaptation, with only the number of variables and their definitions changing. The joint contribution of these two complementary techniques is to refocus the analysis of adaptation from a global concept, or from a developed nations vs. developing nations concept, to a performance analysis specific to each individual nation. Importantly, these techniques identify leading and lagging nations, and quantify the three adaptation gaps for each lagging nation. Finally, they provide a rigorous foundation for a nation-focused investigation into the double inequity of composite adaptation to climate change.

## 6. Results and Discussion

This Section summarises the main findings of the study. Section 6.1 discusses findings based on DEA and Section 6.2 discusses findings based on dominance analysis. Section 6.3 summarises the findings on the inequity of the national distribution of adaptation performance and reinforces the assertions of Stern and the IPCC concerning the double inequity of national adaptation.

### 6.1. DEA Results

Findings from the application of DEA are summarised in Tables 2–4. DEA is first used to aggregate the nine adaptive capacity indicators into an adaptive capacity index ACI and to aggregate the seven adaptation readiness indicators into an adaptation readiness index ARI. This procedure identifies leading and lagging nations in adaptive capacity and adaptation readiness, respectively. The two indices are then combined to generate a composite adaptation index CAI in two ways, by calculating the geometric mean of ACI and ARI, and by applying DEA to aggregate ACI and ARI. Both procedures identify leaders and laggards in adaptation performance, or the ability to enhance adaptive capacity with a supportive institutional environment. The first has the virtue of simplicity, but implicitly treats the two components as being equally important. The second yields information on nations' comparative advantage in adaptation and readiness. The rank correlation between the two composite adaptation indices is calculated to test the concordance of the two strategies.

**Table 2.** Adaptive Capacity Indices ACI.

| Adaptive Capacity Indices | | | |
|---|---|---|---|
| **Leaders** | | **Laggards** | |
| **Nation** | **ACI** | **Nation** | **ACI** |
| | | Cote d'Ivoire | 0.752 |
| | | Guinea-Bissau | 0.752 |
| | | Madagascar | 0.735 |
| | | Libya | 0.73 |
| | | Myanmar | 0.73 |
| | | Pakistan | 0.725 |
| | | Nigeria | 0.719 |
| | | Namibia | 0.709 |
| | | Ethiopia | 0.704 |
| | | Botswana | 0.699 |
| 81 nations | 1 | Congo | 0.685 |
| | | Senegal | 0.671 |
| | | Burkina Faso | 0.658 |
| | | Mali | 0.633 |
| | | Guinea | 0.621 |
| | | Yemen | 0.621 |
| | | Sudan | 0.599 |
| | | Papua New G | 0.588 |
| | | Mauritania | 0.529 |
| | | Eritrea | 0.495 |
| | | Niger | 0.422 |
| mean | 1 | mean | 0.656 |

**Table 3.** Adaptation Readiness Indices ARI.

| Adaptation Readiness Indices | | | |
|---|---|---|---|
| **Leaders** | | **Laggards** | |
| **Nation** | **ARI** | **Nation** | **ARI** |
| Australia | 1 | Eritrea | 0.521 |
| Denmark | 1 | Mozambique | 0.518 |
| Finland | 1 | Togo | 0.508 |
| France | 1 | Papua New G | 0.505 |
| Greece | 1 | Burkina Faso | 0.5 |
| Iceland | 1 | Guinea | 0.5 |
| Korea, Republic of | 1 | Yemen | 0.493 |
| Luxembourg | 1 | Congo | 0.488 |
| Netherlands | 1 | Bangladesh | 0.483 |
| New Zealand | 1 | Zimbabwe | 0.481 |
| Norway | 1 | Nicaragua | 0.472 |
| Sweden | 1 | Ethiopia | 0.465 |
| Switzerland | 1 | Sudan | 0.465 |
| United Kingdom | 1 | Mali | 0.459 |
| Germany | 0.99 | Niger | 0.45 |
| Austria | 0.98 | Pakistan | 0.442 |
| Canada | 0.98 | Cameroon | 0.439 |
| Georgia | 0.962 | Myanmar | 0.435 |
| Belgium | 0.952 | Nigeria | 0.395 |
| Estonia | 0.943 | Afghanistan | 0.382 |
| mean | 0.99 | mean | 0.47 |

**Table 4.** Composite Adaptation Indices CAI.

| Composite Adaptation Indices | | | |
| --- | --- | --- | --- |
| Leaders | | Laggards | |
| Nation | CAI | Nation | CAI |
| Australia | 1 | Guinea-Bissau | 0.645 |
| Denmark | 1 | Bangladesh | 0.643 |
| Finland | 1 | Togo | 0.635 |
| France | 1 | Cote d'Ivoire | 0.632 |
| Greece | 1 | Afghanistan | 0.615 |
| Iceland | 1 | Cameroon | 0.602 |
| Korea, Republic of | 1 | Congo | 0.578 |
| Luxembourg | 1 | Burkina Faso | 0.574 |
| Netherlands | 1 | Ethiopia | 0.572 |
| New Zealand | 1 | Pakistan | 0.566 |
| Norway | 1 | Myanmar | 0.563 |
| Sweden | 1 | Mauritania | 0.56 |
| Switzerland | 1 | Guinea | 0.557 |
| United Kingdom | 1 | Papua New G | 0.553 |
| Germany | 0.995 | Yemen | 0.553 |
| Austria | 0.99 | Mali | 0.539 |
| Canada | 0.99 | Nigeria | 0.533 |
| Georgia | 0.981 | Sudan | 0.528 |
| Belgium | 0.976 | Eritrea | 0.508 |
| Estonia | 0.971 | Niger | 0.436 |
| United States | 0.971 | | |
| mean | 0.994 | mean | 0.569 |

Table 2 lists leaders and laggards in adaptive capacity by presenting the 20 most capable and the 20 least capable nations, and Table 3 lists leaders and laggards in adaptation readiness in the same format. The large number of nations having adaptive capacity index and ARI values of unity illustrates the curse of dimensionality, the difficulty of distinguishing among the leading nations when the number of choice variables is large. Some nations achieve ACI = 1 or ARI = 1 by performing well on several dimensions, while others achieve the same result through specialisation, by excelling on one dimension and lagging on other dimensions, simply by being different. The majority of the 20 most ready nations are European nations and their Western Offshoots, and all these nations are among the 81 most capable nations. (The term "Western Offshoots" was introduced by Maddison [68] to categorise the US, Canada, Australia, and New Zealand.) The curse of dimensionality disappears at the bottom of the rankings for the least capable and least ready nations, the majority of which appear among the UN Least Developed Countries (LDCs). Many are African, most of them sub-Saharan, some are South Asian, and 13 nations appear on both laggard lists. For these nations the advantage of having the freedom to choose weights is offset by the disadvantage of having relatively small values of adaptive capacity and adaptation readiness indicators to which weights are attached. The mean adaptive capacity of laggard nations is barely 65% that of leader nations, and their mean adaptation readiness is even lower, at 47% that of leader nations.

Endogenous weights assigned to leaders differ from those assigned to laggards. Heterogeneity prevails, reflecting nations' varying circumstances and providing a dramatic indication of the value of allowing endogenous weights rather than the more popular fixed weights, although two tendencies appear. Weights attached to leaders' health, ecosystem services and human habitat indicators reflect their relative adaptation strengths, and those attached to laggards' food and infrastructure indicators reflect their relative adaptation strengths. Weights attached to leaders' economic and governance readiness indicators reflect their relative readiness strengths, and those attached to laggards' governance and social readiness indicators reflect their relative readiness strengths.

Table 4 combines adaptive capacity and adaptation readiness by reporting 21 leading nations and 20 lagging nations in composite adaptation, using the geometric mean of adaptive capacity and adaptation readiness indices to generate a CAI. The curse of dimensionality reappears for the most capable nations, 19 of which are European nations or their Western Offshoots. Most of the least capable nations are LDCs, primarily sub-Saharan African, South Asian, and SIDS, and several SIDS such as Kiribati and Tonga are not in the data set. The laggards' mean CAI value is barely half, 57%, the mean CAI value of the leaders. The picture that emerges is one of European nations and their Western Offshoots being institutionally prepared to exploit their relatively abundant adaptive capacities, and LDCs, primarily African nations, lacking the economic, governance and social readiness to exploit their limited adaptive capacities. All 134 nations are mapped according to their composite adaptive capacity index CAI in colour-coordinated Figure 2, with white gaps indicating nations with insufficient data to be included in the exercise. The best performing nations are located at higher latitudes in the northern and southern hemispheres, and the worst performing nations are located in Africa and South Asia at lower latitudes closer to the equator. White gaps indicate nations not among the 134 nations in the data set due to insufficient data.

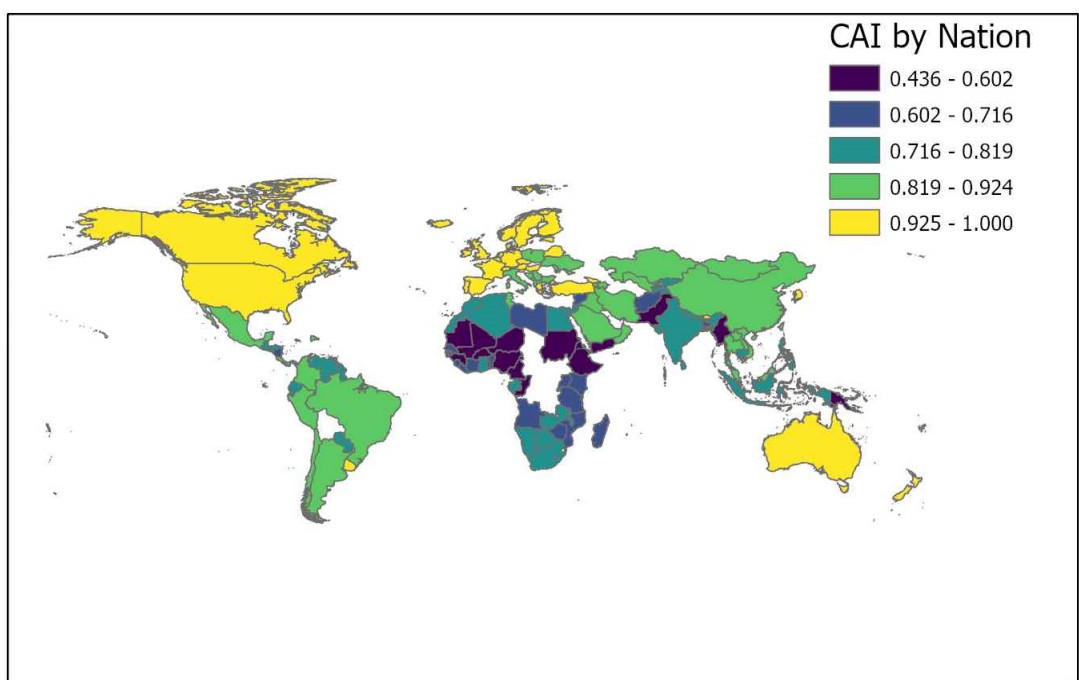

**Figure 2.** Composite Adaptation Indices CAI by Nation.

Results of using DEA to construct a CAI and to identify leaders and laggards are not reported because they are very similar to those using the geometric mean to construct a CAI, with a rank correlation between the two composite adaptation indices of 0.843. A virtue of using DEA to construct a CAI is that, unlike the geometric mean, which weights the two component indices equally, DEA assigns endogenous weights to nations that vary with their circumstances and their relative endowments of adaptive capacity and adaptation readiness in constructing their CAI. A huge majority, 128 of 134 nations, assign zero weights to adaptation readiness, suggesting that most nations, rich and poor, lack the institutional framework that constitutes adaptation readiness, to complement their adaptive capacities. Adom and Amoani [29] and Arezki [69] have emphasised the lack of adaptation readiness in Africa, whose nations dominate the CAI laggards, citing limited climate finance absorptive capacity stemming from relatively weak state capacity, inadequate economic governance, weak financial systems, and inefficient transport systems.

*6.2. Dominance Analysis Results*

Findings from the application of dominance analysis are summarised in Tables 5 and 6. The results of dominance analysis are collected in Table 5 for adaptive capacity and in Table 6 for adaptation readiness. The dominance relationship is demanding, requiring a nation to dominate, or be dominated, by another nation for every indicator, nine for adaptive capacity and seven for adaptation readiness. Nonetheless, empirical dominance relationships are numerous, particularly for adaptation readiness. As with the results in Tables 2 and 3, the majority of the most dominating nations are European nations and their Western Offshoots, and the majority of the most frequently dominated nations are LDCs, most of them African or South Asian. All 20 of the most dominating nations in adaptive capacity in Table 5 appear among the 81 most capable nations in adaptive capacity in Table 2, suggesting a concordance between capability and dominance among leading nations. However, just 12 of the 20 most frequently dominated nations in adaptive capacity are also among the 20 least capable nations in adaptive capacity, suggesting a mild dissonance between capability and dominance among laggard nations.

**Table 5.** Adaptive Capacity Dominance Analysis.

| Adaptive Capacity Dominance | | | |
|---|---|---|---|
| Leaders | | Laggards | |
| **Nation** | **# Dominates** | **Nation** | **# Dominated by** |
| New Zealand | 55 | Yemen | 62 |
| Norway | 51 | Eritrea | 60 |
| Iceland | 45 | Congo | 58 |
| Portugal | 45 | Cambodia | 54 |
| United States | 43 | Niger | 42 |
| Greece | 42 | Sierra Leone | 41 |
| Spain | 42 | Benin | 39 |
| Switzerland | 41 | Madagascar | 37 |
| Netherlands | 40 | Mauritania | 36 |
| Turkey | 35 | Afghanistan | 34 |
| Australia | 34 | Guinea | 34 |
| Canada | 34 | Angola | 33 |
| Austria | 32 | Ethiopia | 33 |
| Panama | 32 | Liberia | 33 |
| Finland | 29 | Sudan | 33 |
| Chile | 28 | Togo | 33 |
| Mexico | 25 | Mali | 32 |
| Sweden | 25 | Bangladesh | 31 |
| Argentina | 23 | Senegal | 27 |
| Italy | 23 | Namibia | 26 |

The number of dominance relationships for adaptation readiness is nearly double the number for adaptive capacity, reflecting the smaller number of indicators on which to dominate or be dominated, and the concordance between capability and dominance is weaker with adaptation readiness than with adaptive capacity. Among the leaders, four of the most dominating nations (Ireland, Portugal, United States and Slovenia) are not among the most ready nations in Table 3, and five of the most ready nations (Greece, Iceland, Luxembourg, Georgia and Belgium) are not among the most dominating nations in Table 6. A similar pattern occurs among the laggards, with three of the most frequently dominated nations (Guinea-Bissau, Angola and Madagascar) not among the least ready nations, and three of the least ready nations (Togo, Burkina Faso and Ethiopia) not among the most frequently dominated nations.

**Table 6.** Adaptation Readiness Dominance Analysis.

| Adaptation Readiness Dominance | | | |
|---|---|---|---|
| **Leaders** | | **Laggards** | |
| **Nation** | **# Dominates** | **Nation** | **# Dominated by** |
| New Zealand | 110 | Afghanistan | 99 |
| Norway | 110 | Nigeria | 89 |
| Australia | 109 | Guinea-Bissau | 86 |
| Iceland | 104 | Zimbabwe | 86 |
| Denmark | 103 | Congo | 85 |
| Sweden | 102 | Eritrea | 83 |
| Finland | 100 | Cameroon | 82 |
| Netherlands | 96 | Pakistan | 79 |
| Korea, Republic of | 94 | Mali | 78 |
| Austria | 93 | Guinea | 73 |
| Estonia | 92 | Nicaragua | 72 |
| Ireland | 91 | Yemen | 72 |
| Canada | 90 | Mozambique | 71 |
| Switzerland | 89 | Sudan | 71 |
| Portugal | 86 | Papua New G | 68 |
| Germany | 85 | Bangladesh | 67 |
| United States | 85 | Myanmar | 67 |
| United Kingdom | 84 | Niger | 64 |
| Slovenia | 83 | Angola | 63 |
| France | 77 | Madagascar | 63 |

Tables 5 and 6 illustrate an important feature of dominance analysis. A nation can dominate often without being among the best (e.g., Portugal in adaptation readiness), and conversely a nation can be among the best without being very dominant (e.g., Greece in adaptation readiness). At the opposite end of the distribution, a nation can be dominated by many other nations without being among the worst performing nations (e.g., Cambodia in adaptive capacity), and conversely a nation can be among the worst performing nations without being dominated by many other nations (e.g., Cote d'Ivoire in adaptive capacity). For leaders and laggards alike, the former outcome occurs when a nation has a similar mix of indicators to many other nations, and the latter outcome occurs when a nation has an unusual mix of indicators.

A second important feature of dominance analysis is the information it provides when one or more indicator is deleted from a dominance relationship. In the case of adaptive capacity dominance, leaders are most affected by deletion of the water indicator, which increases the number of nations they dominate, often by large magnitudes, suggesting that leaders are relatively lacking in the water indicator. Laggards are most affected by the deletion of the ecosystem services indicators, which increases the number of nations that dominate them, suggesting that laggards are relatively well endowed with ecosystem services. In the case of adaptation readiness dominance, most of the leaders are only marginally affected by the deletion of the economic readiness indicator or the governance readiness indicators. However, the number of countries they dominate when the social readiness indicators are deleted increases substantially, suggesting that the leaders are weakest in the social readiness category. Korea is the lone exception, benefiting most from the deletion of the governance readiness indicators. Among the dominance laggards, most are only marginally affected by the deletion of the economic readiness indicator or the social readiness indicators, but the number of dominating countries increases substantially when the governance readiness indicators are deleted, suggesting that the laggards are relatively capable in the governance readiness category.

It is worth noting that New Zealand is a high performer, ranking among the leaders in composite adaptation and the leading nation in both types of dominance. This strong showing is consistent with the findings of King and Jones [70], who augmented the ND-

GAIN data in Table 1 with three additional indicators: arable land availability, renewable energy availability, and isolation. They found New Zealand to have the most favourable "starting conditions" to form a "node of increasing complexity", followed by Iceland, the United Kingdom, Australia, and Ireland. It should be noted that their third additional indicator, isolation, favours island nations.

*6.3. Inequity Results*

Table 7 highlights one dimension of the inequity of national composite adaptation, by listing the GDP per capita of the most and least capable nations ranked by CAI. (GDP per capita data are 2019 GDP per capita PPP (current international $) from the World Bank (https://data.worldbank.org/indicator/NY.GDP.PCAP.PP.CD, accessed on 12 October 2022). The most capable nations have mean CAI 75% greater than that of the least capable nations and have mean GDP per capita nearly 15 times that of the least capable nations. Developing nations cannot afford to invest in composite adaptation. Exceptions are rare; Georgia and Greece are among the composite adaptation leaders despite having relatively low levels of GDP/capita. This finding is consistent with assertions in IPCC Assessment Reports that adaptive capacity is a function of several factors, the first being wealth. It strongly supports the calls of António Guterres for an increase in climate finance and a greatly expanded transfer of this increase from developed nations and international development banks to developing nations, and for a reallocation of the increased funding from mitigation to adaptation.

**Table 7.** The Inequity of Composite Adaptation.

| Composite Adaptation and GDP/Capita | | | | | |
|---|---|---|---|---|---|
| Leaders | | | Laggards | | |
| **Nation** | **CAI** | **GDP/Capita** | **Nation** | **CAI** | **GDP/Capita** |
| Australia | 1 | 52,031 | Guinea-Bissau | 0.645 | 2021 |
| Denmark | 1 | 59,897 | Bangladesh | 0.643 | 4955 |
| Finland | 1 | 51,521 | Togo | 0.635 | 2212 |
| France | 1 | 49,620 | Cote d'Ivoire | 0.632 | 5433 |
| Greece | 1 | 30,842 | Afghanistan | 0.615 | 2152 |
| Iceland | 1 | 60,133 | Cameroon | 0.602 | 3901 |
| Korea, Rep of | 1 | 42,849 | Congo | 0.578 | 3987 |
| Luxembourg | 1 | 119,416 | Burkina Faso | 0.574 | 2268 |
| Netherlands | 1 | 59,675 | Ethiopia | 0.572 | 2315 |
| New Zealand | 1 | 45,073 | Pakistan | 0.566 | 4896 |
| Norway | 1 | 68,345 | Myanmar | 0.563 | 4940 |
| Sweden | 1 | 55,338 | Mauritania | 0.560 | 5566 |
| Switzerland | 1 | 73,144 | Guinea | 0.557 | 2675 |
| United Kingdom | 1 | 49,344 | Papua New G | 0.553 | 4475 |
| Germany | 0.995 | 56,285 | Yemen | 0.553 | 3689 * |
| Austria | 0.990 | 58,641 | Mali | 0.539 | 2420 |
| Canada | 0.990 | 50,661 | Nigeria | 0.533 | 5353 |
| Georgia | 0.981 | 15,623 | Sudan | 0.528 | 4363 |
| Belgium | 0.976 | 54,918 | Eritrea | 0.508 | 1626 * |
| Estonia | 0.971 | 38,294 | Niger | 0.436 | 1276 |
| United States | 0.971 | 65,280 | | | |
| *: 2011 and 2013, the latest years available | | | | | |
| mean | 0.994 | 52,294 | mean | 0.569 | 3623 |

Table 8 reinforces the inequity of national composite adaptation by shifting attention from an income dimension to a responsibility dimension. The most and least capable nations by CAI are compared according to their greenhouse gas emissions per capita

(GHG/capita). (Greenhouse gas emissions per capita data are for 2016 sourced from Our World in Data (https://ourworldindata.org/co2-and-other-greenhouse-gas-emissions, accessed on 12 October 2022). The most capable nations are also the main source of global greenhouse gas emissions, emitting nearly 3.5 times as much per capita as the least capable nations. Again, exceptions are rare; Norway, Sweden and Georgia are among the composite adaptation leaders despite having relatively low levels of GHG/capita. Developing nations are not the source of climate change impacts that threaten them. Taken together, Tables 7 and 8 provide a strong confirmation of Stern's [6] double inequity.

**Table 8.** The Further Inequity of Composite Adaptation.

| Composite Adaptation and GHG/Capita | | | | | |
|---|---|---|---|---|---|
| Leaders | | | Laggards | | |
| Nation | CAI | GHG/Capita | Nation | CAI | GHG/Capita |
| Australia | 1 | 21.39 | Guinea-Bissau | 0.645 | 2.36 |
| Denmark | 1 | 8.17 | Bangladesh | 0.643 | 1.33 |
| Finland | 1 | 11.49 | Togo | 0.635 | 2.05 |
| France | 1 | 5.10 | Cote d'Ivoire | 0.632 | 1.31 |
| Greece | 1 | 8.14 | Afghanistan | 0.615 | 2.73 |
| Iceland | 1 | 9.61 | Cameroon | 0.602 | 8.71 |
| Korea, Republic of | 1 | 12.89 | Congo | 0.578 | 9.99 |
| Luxembourg | 1 | 16.87 | Burkina Faso | 0.574 | 2.07 |
| Netherlands | 1 | 11.01 | Ethiopia | 0.572 | 1.82 |
| New Zealand | 1 | 13.55 | Pakistan | 0.566 | 1.98 |
| Norway | 1 | 4.53 | Myanmar | 0.563 | 4.14 |
| Sweden | 1 | 4.70 | Mauritania | 0.560 | 2.74 |
| Switzerland | 1 | 5.58 | Guinea | 0.557 | 3.89 |
| United Kingdom | 1 | 6.96 | Papua New G | 0.553 | 7.72 |
| Germany | 0.995 | 9.84 | Yemen | 0.553 | 0.87 |
| Austria | 0.990 | 8.21 | Mali | 0.539 | 2.64 |
| Canada | 0.990 | 21.42 | Nigeria | 0.533 | 2.59 |
| Georgia | 0.981 | 4.29 | Sudan | 0.528 | 3.81 |
| Belgium | 0.976 | 9.46 | Eritrea | 0.508 | 2.36 |
| Estonia | 0.971 | 15.48 | Niger | 0.436 | 2.05 |
| United States | 0.971 | 18.06 | | | |
| mean | 0.994 | 11.64 | mean | 0.569 | 3.36 |

Table 9 combines income and responsibility to provide a holistic confirmation of Stern's double inequity of adaptation performance. The most and least capable nations by CAI are compared according to their generic inequity index GII, constructed as the geometric mean of their income and responsibility indices GDP per capita and GHG per capita. Laggards have mean CAI 57% of that of leaders, and a mean GII 14% of that of leaders. Those nations most capable of adapting to climate change are both wealthy and the source of most causal greenhouse gas emissions. If laggards and leaders are defined more generously as the bottom and top 50 nations based on CAI, the magnitude of the double inequity is barely dented. The mean CAI of redefined laggards rises to 69% of that of redefined leaders, and their mean GII is 26% of that of redefined leaders.

**Table 9.** The Generic Inequity of Composite Adaptation.

| Composite Adaptation and Generic Inequity | | | | | |
|---|---|---|---|---|---|
| **Leaders** | | | **Laggards** | | |
| **Nation** | **CAI** | **GII** | **Nation** | **CAI** | **GII** |
| Australia | 1 | 33.364 | Guinea-Bissau | 0.645 | 2.185 |
| Denmark | 1 | 22.121 | Bangladesh | 0.643 | 2.564 |
| Finland | 1 | 24.329 | Togo | 0.635 | 2.128 |
| France | 1 | 15.903 | Cote d'Ivoire | 0.632 | 2.666 |
| Greece | 1 | 15.841 | Afghanistan | 0.615 | 2.424 |
| Iceland | 1 | 24.037 | Cameroon | 0.602 | 5.829 |
| Korea, Republic of | 1 | 23.505 | Congo | 0.578 | 6.312 |
| Luxembourg | 1 | 44.889 | Burkina Faso | 0.574 | 2.168 |
| Netherlands | 1 | 25.634 | Ethiopia | 0.572 | 2.053 |
| New Zealand | 1 | 24.713 | Pakistan | 0.566 | 3.115 |
| Norway | 1 | 17.603 | Myanmar | 0.563 | 4.522 |
| Sweden | 1 | 16.127 | Mauritania | 0.560 | 3.905 |
| Switzerland | 1 | 20.193 | Guinea | 0.557 | 3.224 |
| United Kingdom | 1 | 18.533 | Papua New G | 0.553 | 5.877 |
| Germany | 0.995 | 23.533 | Yemen | 0.553 | 1.789 |
| Austria | 0.990 | 21.947 | Mali | 0.539 | 2.529 |
| Canada | 0.990 | 32.941 | Nigeria | 0.533 | 3.721 |
| Georgia | 0.981 | 8.188 | Sudan | 0.528 | 4.075 |
| Belgium | 0.976 | 22.787 | Eritrea | 0.508 | 1.957 |
| Estonia | 0.971 | 24.344 | Niger | 0.436 | 1.619 |
| United States | 0.971 | 34.335 | | | |
| mean | 0.994 | 23.565 | mean | 0.569 | 3.233 |

The double inequity in Table 9 is confined to leaders and laggards, but the double inequity affects all nations, with a strong positive correlation between nations' CAI and their GII of 0.684. To illustrate the entire distribution rather than just its upper and lower tails, GII indices for 131 nations are mapped in colour-coordinated Figure 3, with the same two white gaps. (Three nations are deleted in constructing the generic inequity index GII. GDP per capita data are unavailable for Syria, and Bhutan and Gabon report negative greenhouse gas emissions. For explanations for Bhutan's negative emissions see https://ourworldindata.org/co2/country/bhutan, accessed on 12 October 2022 and for Gabon's see https://ourworldindata.org/co2/country/gabon, accessed on 12 October 2022). With few exceptions, the wealthy source nations are European nations and their Western Offshoots, and the poor non-source nations are located in Africa, the Sub-Continent, and South Asia. A comparison of Figure 3, which maps GII, with Figure 2, which maps CAI, provides a vivid depiction of Stern's double inequity. With a few notable exceptions mentioned above, the two maps are nearly indistinguishable.

Tables 8 and 9, and Figures 2 and 3, have a geographical interpretation as well as an inequity interpretation. Composite adaptation leaders are relatively rich and largely responsible for climate change impacts and are located at higher latitudes in the northern and southern hemispheres (e.g., Canada in the north and New Zealand in the south). Composite adaptation laggards are relatively poor and not responsible, and cluster at lower latitudes close to the equator (e.g., Togo and Papua New Guinea). This geographical interpretation was proposed by Nordhaus [71], who compared GDP per capita with latitude and temperature for a sample of 77 nations. He found rich nations located in cool latitudes away from the equator and poor nations located in warm latitudes near the equator. (Nordhaus was co-recipient of the 2018 Nobel Prize in Economic Sciences "for integrating climate change into long-run macroeconomic analysis". In Nordhaus [72], he originally

proposed a global warming target of 2 °C above pre-industrial levels now enshrined in the Paris Agreement.)

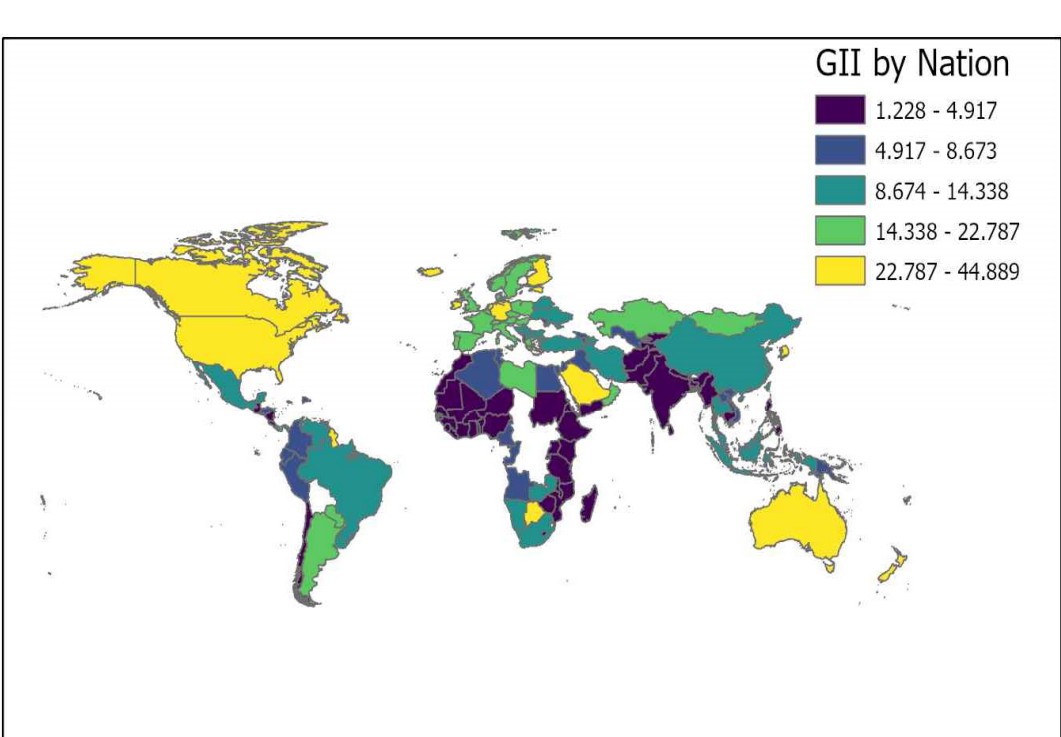

**Figure 3.** Generic Inequity Indices GII by Nation.

### 7. Conclusions

The introduction set three objectives for this study: to create a database of indicators conforming to the IPCC concept of adaptation, to propose analytical techniques with which to assess the adaptation performance of nations, and to explore the distribution of composite adaptation performance among nations and empirically assess its inequities.

The database is created in Section 4 and incorporates the adaptive capacity and adaptation readiness indicators proposed in successive IPCC Assessment Reports. These indicators reflect a belief that a supportive institutional environment provides the readiness essential to the success of adaptation efforts. The database is drawn from the ND-GAIN database, although it is not equivalent to it, and it satisfies the essence of the Ford and Berrang-Ford [56] requirements for successful adaptation tracking.

In Section 5 a linear programming distance to frontier technique, DEA, is augmented with a dominance analysis, providing complementary insights into nations' composite adaptation performance, by identifying leaders and laggards according to different criteria, and by identifying the indicators at which they perform relatively well or relatively poorly. Dominance analysis adds value to DEA by dispensing with the frontier concept and evaluating nations' adaptation performance relative to other nations, rather than relative to an adaptation frontier. Together the two techniques provide a rigorous analytical foundation for subsequent empirical analysis of the composite adaptation performance of nations.

The third objective is achieved in two stages. The empirical analysis in Section 6 identifies leading and lagging nations in terms of their relative adaptive capacity and their relative adaptation readiness separately, and in terms of their relative composite adaptation performance. The overriding impression gained is one of very large dispersion in nations' adaptation performance. The composite adaptation gap between leading and lagging nations is large, with lagging nations' adaptation performance on the order of 57% of that of leading nations. The gap is attributable primarily to inadequate adaptation readiness of

institutional environments that plagues nearly all nations and is particularly severe among lagging nations. This impression of large dispersion is reinforced when DEA is used to aggregate adaptive capacity and adaptation readiness, with most nations weighting the former more heavily than the latter. When the distance to frontier analysis is augmented with a dominance analysis on adaptive capacity and adaptation readiness criteria separately, the significance of adaptation readiness is strengthened. Dominance relationships are roughly twice as frequent with adaptation readiness as with adaptive capacity, attesting further to the importance of a supportive institutional environment. These findings highlight the empirical significance of the complementarity between adaptive capacity and adaptation readiness, and quantify the magnitudes of the three adaptation gaps, two results that have received insufficient attention in the literature.

In the second stage of the empirical analysis composite adaptation leaders and laggards are identified geographically. In terms of both distance to frontier and dominance analyses, composite adaptation leaders are overwhelmingly European nations and their Western Offshoots located in higher latitudes in the northern and southern hemispheres, and laggards are equally overwhelmingly least developed countries, most of them sub-Saharan African and South Asian, located in lower latitudes close to the equator. The distance to the equator principle of economic development applies equally well to climate change adaptation performance.

When an income dimension is added to the characterisation, leaders have approximately 15 times the GDP per capita as laggards have. This relationship applies to the entire distribution of nations, not just to the leading and lagging tails; the correlation between income and composite adaptation performance is 0.75. National composite adaptation performance varies positively and strongly with national income, as the IPCC asserts. This finding illustrates one of Stern's double inequities of adaptation; the poorest nations are the least able to adapt to climate change impacts.

When responsibility for climate change is added to the characterisation, leaders generate more than three times the amount of GHG emissions per capita as laggards do. This relationship also holds for the entire distribution of nations; national composite adaptation performance varies positively, although not strongly due to a few prominent outliers, with responsibility for climate change. This finding illustrates the other of Stern's double inequities of adaptation; nations least responsible for causal greenhouse gas emissions are least able to adapt to their impacts.

When a combination of income and responsibility is added to the characterisation, Stern's double inequity is clearly revealed. The correlation between a combination of income and responsibility and composite adaptation performance is 0.68. National income and responsibility for climate change vary positively and strongly with composite adaptation performance. Those nations having weak composite adaptation lack the resources to adapt to climate change attributable largely to those nations having relatively abundant composite adaptation. Stern's double inequity is portrayed graphically in Figures 2 and 3, which are barely distinguishable. These findings provide analytically based empirical results confirming the well-known but inadequately documented double inequity of climate change. They quantify each inequity gap, and a generic double inequity gap, for each nation. Each of these gaps is large on average, and enormous for some nations.

These findings also reveal two limitations of the research. One is illustrated by the white gaps in Figures 2 and 3 representing Bolivia in the western Amazon basin and the Democratic Republic of Congo and other nations in central Africa. These regions contain major portions of the two largest rainforests in the world. Given the importance of the ecosystems in these two regions, and their vulnerability to climate change, it would have been desirable to include these nations in the empirical analysis. Despite our efforts to retain as many nations as possible, insufficient data are available for these nations to allow their inclusion. A second limitation involves the scope of adaptation. Although the underlying data provide an adequate basis for assessing adaptation in the human environment, they provide a limited basis for assessing adaptation in the natural/ecological

environment. The International Union for the Conservation of Nature (IUCN) has conducted studies and amassed data on ecosystem-based adaptation that complement our knowledge of adaptation focused on the human environment and expand the elements related to natural/ecosystem adaptation to climate change. (See, for example, IUCN [73] and Keith et al. [74] for details on adaptation in the natural/ecosystem environment.) It would be worthwhile in subsequent research to determine if it is possible to merge the IUCN natural/ecosystem adaptation data with the ND-GAIN largely human adaptation data to gain a more complete picture of adaptation.

A lively literature has emerged that regards climate change as a justice issue. Although it is not among the 17 United Nations Sustainable Development Goals, climate justice " . . . looks at the climate crisis through a human rights lens . . . " (https://www.un.org/sustainabledevelopment/blog/2019/05/climate-justice/, accessed on 15 October 2022), thereby providing a holistic but loosely defined notion of the (in)ability to achieve these goals. This study has addressed climate change as an equity issue by providing a rigorous analytically based confirmation of Stern's double inequity assertion, a positive assertion that can be and has been tested empirically against a measurable alternative of adaptation equality. This empirical approach to climate change as an equity issue contrasts with the popular assertion that treats climate change as a justice issue. The latter is a normative assertion that can be debated but cannot be tested empirically until a benchmark is developed against which climate justice can be measured.

**Author Contributions:** Conceptualization: H.K.E., C.A.K.L. and J.E.L.; methodology: C.A.K.L.; software: H.K.E. and J.E.L.; validation: H.K.E., C.A.K.L. and J.E.L..; formal analysis: C.A.K.L. and J.E.L.; investigation C.A.K.L. and J.E.L.; resources: H.K.E., C.A.K.L. and J.E.L.; resources: H.K.E., C.A.K.L. and J.E.L.; data curation: J.E.L. and C.A.K.L.; writing—original draft preparation: C.A.K.L. and J.E.L.; writing—review and editing: C.A.K.L. and J.E.L.; visualization: H.K.E.; supervision: C.A.K.L.; project administration: H.K.E., C.A.K.L. and J.E.L.; funding acquisition: none. All authors have read and agreed to the published version of the manuscript.

**Funding:** This research received no external funding.

**Institutional Review Board Statement:** Not applicable.

**Informed Consent Statement:** Not applicable.

**Data Availability Statement:** Data used in this study are publicly available in [14].

**Conflicts of Interest:** The authors declare no conflict of interest.

## Appendix A

**Table A1.** Summary Statistics.

| Composite Adaptation Indicators | | | | |
|---|---|---|---|---|
| **Indicator** | **Mean** | **Std Dev** | **Min** | **Max** |
| Adaptive Capacity | | | | |
| Agricultural Capacity | 0.184 | 0.229 | 0 | 1 |
| Child Nutrition | 0.820 | 0.176 | 0.291 | 1 |
| Water | 0.221 | 0.189 | 0.003 | 0.839 |
| Medical Staffs | 0.426 | 0.354 | 0.012 | 1 |
| Protected Biomes | 0.569 | 0.196 | 0.149 | 0.875 |
| International Environmental Conventions | 0.335 | 0.250 | 0 | 1 |
| Trade and Transport Infrastructure | 0.414 | 0.167 | 0.118 | 0.838 |
| Paved Roads | 0.489 | 0.330 | 0.027 | 1 |
| Electricity Access | 0.846 | 0.245 | 0.143 | 1 |
| Adaptation Readiness | | | | |
| Economic Readiness | | | | |
| Doing Business | 0.430 | 0.146 | 0.134 | 0.772 |

**Table A1.** *Cont.*

| Composite Adaptation Indicators | | | | |
| --- | --- | --- | --- | --- |
| **Indicator** | **Mean** | **Std Dev** | **Min** | **Max** |
| Governance Readiness | | | | |
| Political Stability and Non-Violence | 0.531 | 0.159 | 0.094 | 0.855 |
| Control of Corruption | 0.404 | 0.228 | 0.041 | 0.924 |
| Rule of Law | 0.509 | 0.188 | 0.056 | 0.880 |
| Regulatory Quality | 0.490 | 0.191 | 0.056 | 0.906 |
| Social Readiness | | | | |
| ICT Infrastructure | 0.461 | 0.144 | 0.209 | 0.732 |
| Education | 0.286 | 0.211 | 0.004 | 1 |

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
