# Peer review of "The Inequities of National Adaptation to Climate Change"

_resources, doi:10.3390/resources12010001_

Round 1
Reviewer 1 Report
Although the results of this research are not altogether different from what might be intuitive and fairly predictable, the analytical framework and indices represent an important advance in quantitative methods for assessing differential capacity among nations for adaptation to climate change effects, and for monitoring changes in this relative capacity over time.
Use of the ND-GAIN indicators as the basis for data acquisition and structuring is certainly valid, and supports a good approach to assessing the adaptation capacity in the human environment, but it provides a very limited basis for assessing the capacity for protecting biological resources. Participation in international environmental conventions, and protected biomes, convey important information. But this information would be much more meaningful if there was some way to capture the proportion of a nation's land that is in some sort of protected status, preferably with reference to the degree of protection provided along the lines of the IUCN protected area definitions.
It is not entirely clear from the manuscript why large sections of central and southern Africa and sections of the western Amazon basin are left out of the analysis. Given the importance of the biological resources in these regions and their vulnerability to climate change effects, it would be particularly importance to include them in the proposed indices of adaptation capacity and readiness. For the purposes of this manuscript, perhaps a fuller explanation of their absence--and perhaps what would be needed in order to be able to include these areas--would be valuable.
Author Response
Response to reviewer #1
Thank you for your valuable comments on our submission. Our responses follow.
- ND-GAIN indicators are useful for assessing adaptive capacity in the human environment, but less so in the natural/ecosystem environment.
We agree, and we note in the first paragraph of Section 2 that the IPCC “…noted that adaptive capacity is more limited in natural systems than in human systems”. Our data are constructed from the ND-GAIN data base, and so as you note our empirical findings are largely limited to human adaptation, protected biomes being the sole exception.
- Why are large sections of central and southern Africa and sections of the western Amazon left out of the analysis?
In the penultimate paragraph of concluding Section 7 we explain the source of the two white gaps in Figures 2 and 3---ND-GAIN does not have complete data for Bolivia and the Democratic Republic of Congo, which together contain major portions of the two largest rainforests in the world. We also note that it would be worthwhile to determine if it is possible to merge IUCN natural/ecosystem adaptation data with ND-GAIN human adaptation data to gain a more complete picture of total adaptation.
In concluding Section 7 we follow up on your two comments by referring to the work of the IUCN on ecosystem-based adaptation, and in accompanying endnote 15 we refer to IUCN (2022) and Keith et al. (2022).
Reviewer 2 Report
This is a very interesting work. However, there are some points that need clarification.
This work is referred to the inequities of countries and not of citizens. The title and the text can lead to misunderstandings. I suggest to modify the title and the corresponding texts.
The authors use some data, but, as these data are indicators and not raw data, a critique on their validity must be provided.
P13, L600 and next. It would be useful to use specific weights for the adaptive capacity of countries, or at least rank the indicators as all are not equally important.
The main problem is that, according to the data used and the analysis applied, the results are, at least, strange. Greece, a country with several economic issues the last 10 years and small efforts to mitigation/adaptation to climate change, is found very high in the list. Also, other similar countries, such as Georgia and Estonia, rank very high. Page 18 shows several countries, such as Portugal or Spain or Panama, ranking very high. From these results, it is obvious that only calculations based on indicators can lead to biased conclusions. I suggest to compare these results of the national plans of those countries (mainly Greece, Georgia and Estonia) and depict the similarities/differences. Otherwise, the results of this paper can be seen as biased.
Author Response
Response to reviewer #2
Thank you for your valuable comments on our submission. Our responses follow.
- The work refers to adaptation inequities of nations rather than individuals, so change the title of the paper.
We take your point, and we have changed the title of the paper and related text by replacing “Global” with “National”.
- The data are indicators rather than raw data and may not be reliable.
This is correct, although we expect indicators to be as reliable as the raw data from which they are obtained. The transformation procedure is widely used in many disciplines and preserves the important property of monotonicity, which underlies our belief in the reliability of the indicators. We have expanded endnote 6 to alert readers to the fact that University of Notre Dame (2021) provides both raw data and derived indicators, and University of Notre Dame (2015; 6-8) provides details of the transformation procedures.
- At old L600 and following, discuss the endogenous weights nations attach to indicators.
Although we have not added a new endogenous weights table to an already long paper, we have added a discussion of the relative magnitudes of endogenous weights nations attach to their indicators, but not at the location the reviewer suggests because in that location we discuss the advantages of allowing endogeneity. We discuss estimated endogenous weights in the third paragraph of Section 6.1. On average, adaptation leaders attach relatively high weights to health and economic and governance, and adaptation laggards attach relatively high weights to food and social readiness. We also note that average weights conceal large variation across nations, which is a dramatic indication of the value of allowing endogenous weights rather than the more popular fixed weights.
- Some results “are, at least, strange”.
We agreed with this point initially, for leaders although not for laggards, in Tables 2-9. However on further consideration we do not find the rankings of nations such as Estonia, Georgia and Greece in DEA, and Portugal and Spain in dominance, to be “strange”. GDP/capita in these nations lags mean GDP/capita for leaders, often by wide margins, and public finances in some of these nations are in poor shape. But these popular variables are not among the composite adaptation indicators we use. Although adaptation varies positively with income, the correlation is not perfect, and these four nations illustrate this fact. We make this point in the context of composite adaptation in Tables 7 and 8, which compare composite adaptation with GDP/capita and GHG/capita. There Georgia, Greece and Estonia are among composite adaptation leaders despite having relatively low levels of GDP/capita, and Norway, Sweden and Georgia are among the adaptation leaders despite having a relatively low level of GHG/capita. These nations illustrate the fact that correlations between composite adaptation and either income or responsibility are positive but less than unity.
Reviewer 3 Report
This is a very good paper and brings up an interesting topic. However, there are a few concerns that if resolved, paper will be able to be published.
- The citation in lines 132 and 890 is written incorrectly.
Author Response
Response to reviewer #3
Thank you for your valuable comments on our submission. Our responses follow.
- The citations at lines 132 and 890 are incorrect.
You are correct. We have corrected the citation at old line 132 by replacing the United Nations link with a reference to The Global Commission on Adaptation (2019), which we have added to the References. We have corrected the citation at old line 890 by noting that climate justice is not among the United Nations Sustainable Development Goals, by retaining the link to the United Nations, and by modifying the rest of the sentence.
- Include details of the indicators in Table 1.
We have added two sentences immediately above Table 1 to conclude the third paragraph of Section 4 noting that the indicators capture both capacity and access dimensions and providing a reference to University of Notre Dame (2015), which provides descriptions of and rationale for each indicator in Table 1.
- Write more about the theoretical implications and empirical limitations of the research after the discussion.
The theoretical implications of the research are hinted at in the penultimate paragraph of introductory Section 1. In concluding Section 7 we argue that the two analytical techniques we use “provide a rigorous analytical foundation” for the empirical analysis we conduct.
The empirical analysis makes three contributions, which we also note in Section 7. First, we demonstrate that the two techniques provide complementary information, in the sense that each adds value to the other. Second, the techniques enable us to quantify adaptation gaps for each nation. Third, and most significantly, the techniques clarify and quantify the double inequity of national climate change adaptation.
In concluding Section 7 we also note two empirical limitations of the research. First, the ND-GAIN data base does not have complete information for all nations, and this has forced us to delete some nations from our empirical analysis. Another reviewer has pointed to the white gaps in Figures 2 and 3, identifying Bolivia and the Democratic Republic of Congo, home of the two largest rainforests in the world, as providing incomplete data and therefore missing from our empirical analysis. The second, and related, limitation of the research is that, while ND-GAIN data provide a good basis for evaluating human adaptation, they provide an inadequate basis for evaluating natural/ecosystem adaptation. Consequently, our findings pertain primarily to human adaptation and have little relevance to natural/ecosystem adaptation. We acknowledge this limitation of the research in concluding Section 7, and in new endnote 15 we have added two references to work of the International Union for the Conservation of Nature. We also note the possibility of merging IUCN and ND-GAIN data to provide a more complete representation of national adaptation to climate change.
Reviewer 4 Report
While the result is potentially useful, it is not clear what the novel contribution of this study is. The introduction and review parts are lengthy, and it seems this paper is just rephrasing the well-recognized fact. At least, the authors should justify why the adopted methods (the distance to frontier analysis and the dominance analysis) are appropriate to measure the adaptation performance of the nations. Although these methods can create an order and rank the nations based on given indicators, they do not assure the crated rank appropriately reflect the actual level of the adaptation performance (just like an unsupervised classification). I recommend re-writing the manuscript, and the contribution to the existing literature should be made clear.
Author Response
Response to reviewer #4
Thank you for your valuable comments on our submission. Our responses follow.
- What is the novel contribution of our study?
The novel contributions of our study are enumerated in the penultimate paragraph of introductory Section 1, and again in concluding Section 7. Briefly, they begin with the creation of a set of indicators that can be combined into a composite adaptation index for a large number of nations. Such an index has not heretofore appeared in the literature. The second novel contribution of our study is to apply a complementary pair of analytical techniques to these data to generate a range of empirical measures of nations’ composite adaptation performance. We believe this contribution is also new. The third novel contribution of our study is to apply these data and these techniques to conduct an empirical evaluation of the proclaimed double inequity of national adaptation to climate change, and to quantify its significance for a large number of nations. Although the double inequity is well established, our empirical quantification of it and each of its two components is new.
- Why are the analytical techniques appropriate?
We assert in Section 5 that these techniques are appropriate when the objective is to quantify empirically various stylized facts about adaptation to climate change. These techniques allow us to identify best practice nations and the gaps between them and other nations. They also allow us to identify dominating and dominated nations and the gaps between them, without recourse to the best practice concept. The appropriateness of these techniques is illustrated throughout Section 6, which contains quantitative findings not available elsewhere in the literature.
- What are the contributions to the existing literature?
Our contributions are asserted in the penultimate paragraph of introductory Section 1, illustrated empirically with a large data set in Section 6, and reiterated in concluding Section 7. Briefly, our merging of indicators of adaptive capacity with indicators of adaptation readiness into an index of composite adaptation is new, both theoretically and empirically. Our subsequent use of composite adaptation to investigate the double inequity of national climate change adaptation is also new, both theoretically and empirically. Our empirical demonstration of the large magnitudes and geographical concentration of the double inequity provides new empirical evidence.
Round 2
Reviewer 2 Report
This work can be published now